# Band insulator to Mott insulator transition in 1T-TaS$_2$

Y. D. Wang[1], W. L. Yao[1], Z. M. Xin[1], T. T. Han [1], Z. G. Wang[1], L. Chen [1], C. Cai[1], Yuan Li [1,2] & Y. Zhang [1,2 ✉]

1T-TaS$_2$ undergoes successive phase transitions upon cooling and eventually enters an insulating state of mysterious origin. Some consider this state to be a band insulator with interlayer stacking order, yet others attribute it to Mott physics that support a quantum spin liquid state. Here, we determine the electronic and structural properties of 1T-TaS$_2$ using angle-resolved photoemission spectroscopy and X-Ray diffraction. At low temperatures, the 2π/2c-periodic band dispersion, along with half-integer-indexed diffraction peaks along the c axis, unambiguously indicates that the ground state of 1T-TaS$_2$ is a band insulator with interlayer dimerization. Upon heating, however, the system undergoes a transition into a Mott insulating state, which only exists in a narrow temperature window. Our results refute the idea of searching for quantum magnetism in 1T-TaS$_2$ only at low temperatures, and highlight the competition between on-site Coulomb repulsion and interlayer hopping as a crucial aspect for understanding the material's electronic properties.

[1] International Center for Quantum Materials, School of Physics, Peking University, 100871 Beijing, China. [2] Collaborative Innovation Center of Quantum Matter, 100871 Beijing, China. ✉email: yzhang85@pku.edu.cn

Transition-metal di-chalcogenides are layered quasi-two-dimensional materials that not only show prominent potentials for making ultra-thin and flexible devices, but also exhibit rich electronic phases with unique properties[1–3]. $1T$-$TaS_2$ is one prominent example. The crystal structure of $1T$-$TaS_2$ consists of S–Ta–S sandwiches, which in turn stack through van der Waals interactions. It is structurally undistorted and electronically metallic at high temperatures. With decreasing temperature, it undergoes successive first-order transitions, resulting in the formation of various charge density waves (CDWs) with increasing degree of commensurability[4–7]. As shown in Fig. 1a, with cooling, $1T$-$TaS_2$ sequentially enters an incommensurate CDW (I-CCDW) phase below 550 K, a nearly commensurate CDW (NC-CDW) phase below 350 K, and finally a commensurate CDW (C-CDW) phase below 180 K. Prominent hysteretic behavior can be observed when comparing the cooling and heating resistivities data. Upon heating, $1T$-$TaS_2$ enters triclinic CDW (T-CDW) phase at 220 K, and then the NC-CDW phase at 280 K. The space modulations of different CDW phases are illustrated in the inset of Fig. 1a. Every adjacent 13 Ta-atoms accumulate together, which is called star-of-David (SD) cluster[4–7]. Within one hexagonal SD, 12 Ta-atoms pair and form 6 occupied bands, leaving the center Ta atom localized and unpaired alone. As a result, the insulating ground state of $1T$-$TaS_2$ has been proposed to be a Mott insulator[8–13]. In the C-CDW phase, the SD clusters cover entire lattice forming commensurate $p\left(\sqrt{13}\times\sqrt{13}\right)R13.9°$ phase. The localized electrons with $S = 1/2$ spin arrange on a triangular lattice, making this system a promising candidate material for realizing quantum spin liquid[14–17].

Distinct from the possible Mott physics that occurs within one $TaS_2$ layer, interlayer coupling has recently been emphasized as a crucial aspect to understand the insulating property of $1T$-$TaS_2$[18–20]. Through an interlayer Peierls CDW transition, interlayer staking order forms in the C-CDW phase. First, every two adjacent $TaS_2$ layers accumulate, forming dimerized bilayer structure. The dimerized $TaS_2$ bilayers then stack onto each other forming different types of staking order with different in-plane sliding configurations[19,20]. Recent scanning tunneling microscopy and transport measurements show possible existence of the stacking order in $1T$-$TaS_2$[21,22]. Band calculations show that without considering the strong electronic correlation the interlayer staking order itself can open a band gap at the Fermi energy ($E_F$), yielding an insulating property of $1T$-$TaS_2$. Under this scenario, the ground state of $1T$-$TaS_2$ is a band insulator rather than a Mott insulator.

The low-energy electronic structure of $1T$-$TaS_2$ consists of one half-filled nonbonding Ta 5d band. In the IC-CDW phase, angle-resolved photoemission spectroscopy (ARPES) data shows that one Ta 5d band disperses crossing $E_F$, forming six oval-like circles around the Brillouin zone boundaries (M) (Fig. 1b). This is well consistent with the first-principle band calculations[11]. When entering the C-CDW phase, the CDW potential folds the Brillouin zone. The Fermi surface reconstructs into multiple disconnected spots, following the $p\left(\sqrt{13}\times\sqrt{13}\right)R13.9°$ periodicity. Band folding and band gap opening split the Ta 5d band into a manifold of narrow subbands. One flat band emerges at ~200 meV below $E_F$ (Fig. 1c). Under the Mott scenario, this flat band was attributed to the lower Hubbard band[8–13]. Previous time-resolved ARPES and inverse-photoemission spectroscopy also reported the observation of Mott gap between the flat band and upper Hubbard band[23–25], supporting the existence of Mott insulating ground state. However, recent ARPES studies show that the flat band is energy dispersive along the out-of-plane momentum ($k_z$) direction[19,26,27], which favors the existence of interlayer stacking order. It is then crucial to make a consensus on the detailed temperature evolution of electronic structure in $1T$-$TaS_2$. The results would help to understand how the on-site Coulomb repulsion and interlayer hopping play roles in $1T$-$TaS_2$.

In this work, we characterize the temperature dependence of electronic and lattice structures of $1T$-$TaS_2$ using ARPES and X-ray diffraction (XRD). We find that, at low temperatures, the

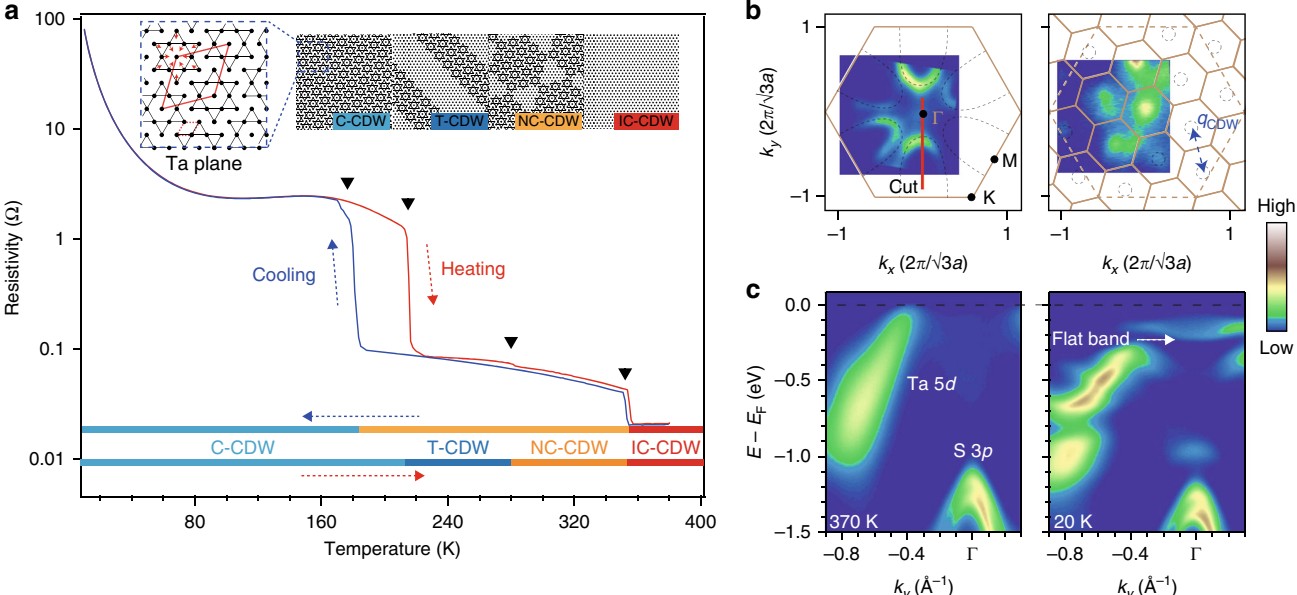

**Fig. 1 Transport property and electronic structure of 1T-TaS₂. a** Temperature dependence of in-plane resistivity taken upon heating and cooling. The black arrows mark the first-order transition points. The bars below the curves indicate temperature intervals corresponding to the different phases. Inset panels in **a** are schematic illustrations for the in-plane lattice modulations in the commensurate CDW (C-CDW) phase, nearly commensurate CDW (NC-CDW) phase, Triclinic CDW (T-CDW) phase, and In-commensurate CDW (IC-CDW) phase, respectively. The SD clusters are marked with the dark outlines. Black dots stand for the Ta atoms. **b** and **c** Fermi surface mappings and corresponding high symmetry cuts taken at 370 and 20 K, respectively. The cut momenta are illustrated as a red line in **b**.

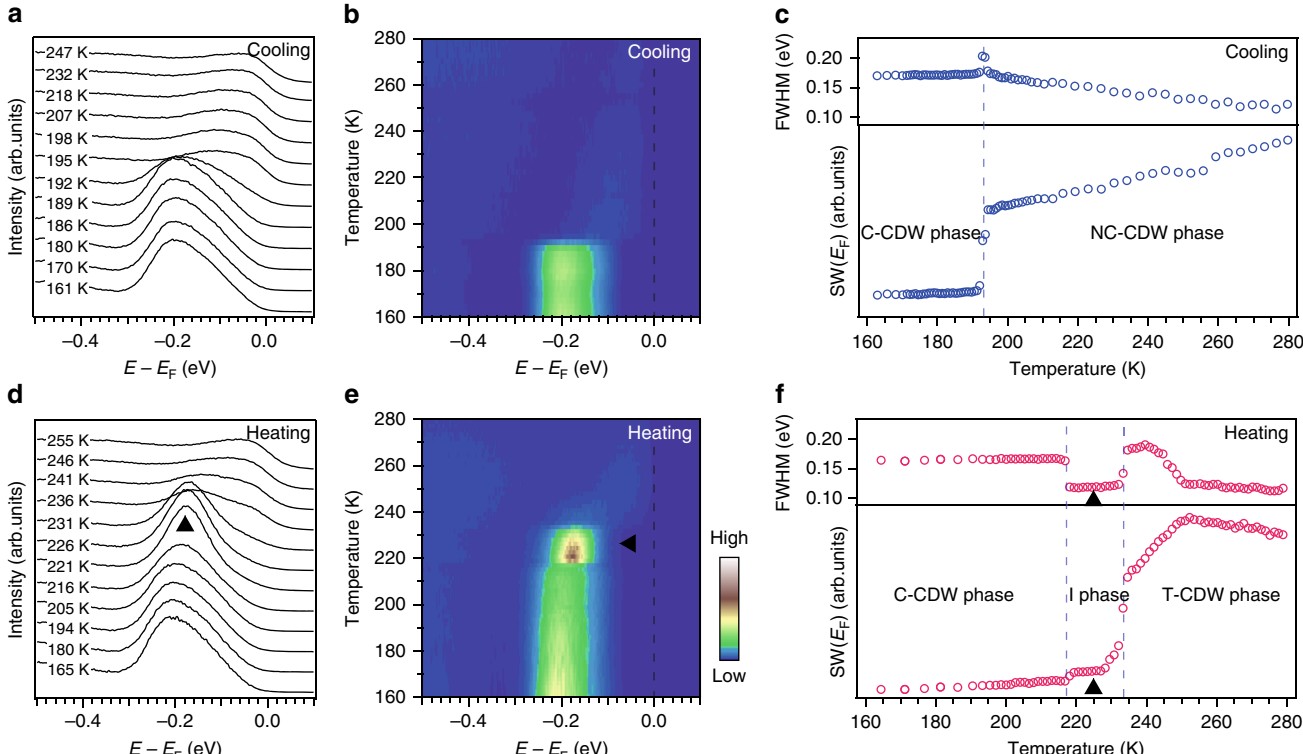

**Fig. 2 Temperature-dependent ARPES study of 1$T$-TaS$_2$ upon cooling and heating. a** Temperature dependence of energy distribution curves (EDCs) taken at the $\Gamma$ point upon cooling. **b** The merged image of the data in **a**. **c** Corresponding temperature dependence of full width at half maximum (FWHM) for the data in **a**, and the spectral weight (SW) taken at where the Ta 5$d$ band crosses $E_F$ ([−0.1, 0.1 eV], [−0.4, −0.3 Å$^{-1}$]). **d**–**f** are the same as **a**–**c** but for the heating process. Black arrows mark the intermediate (I) state.

band dispersion and diffraction peaks show $2\pi/2c$ and $2c$ periodicity, which indicates that the ground state of 1$T$-TaS$_2$ is a band insulator with interlayer dimerization. More intriguingly, at high temperatures, the system undergoes an insulator-to-insulator transition and enters an intermediate Mott insulating state, which only exists in a narrow temperature region. Our results show that the energy scales of in-plane hopping, on-site Coulomb repulsion, and interlayer hopping are all comparable in 1$T$-TaS$_2$. The competition between these interactions are responsible for the complex electronic properties of 1$T$-TaS$_2$.

## Results

**Temperature dependence of electronic structure**. To characterize how the flat band forms in the C-CDW phase, we show the detailed temperature dependence of the energy distribution curves (EDCs) taken at the Brillouin zone center ($\Gamma$) (Fig. 2). Upon cooling, the spectral weight of the flat band shifts from $E_F$ to −200 meV at 193 K, indicating an energy gap opening at $E_F$. In contrast to the cooling process, the temperature evolution of EDCs during the heating process clearly identifies two-phase transitions (Fig. 2d–f). The first phase transition occurs at 217 K, where the peak width narrows and the peak position shifts from −210 to −170 meV. The second phase transition follows at 233 K, where the spectral weight of the flat band shifts to $E_F$.

In comparison with the resistivity data, the gap opening transition at 193 K can be attributed to the NC-CDW to C-CDW transition upon cooling, while the transition at 217 K coincides with the C-CDW to T-CDW transition upon heating. There is a small temperature deviation between the transition temperatures determined by different techniques. This can be explained by the standard deviation of transition temperatures among different samples due to a small sample inhomogeneity

(Supplementary Figs. 1–3). However, the transition at 233 K does not show up in the transport measurements. It also cannot be explained by a sample inhomogeneity, because the temperature-dependent data are well reproducible in many different samples (Supplementary Fig. 1). Our results thereby identify an undiscovered intermediate (I) phase, which exists in a narrow temperature region (217–233 K) at the C-CDW to T-CDW phase transition. For the transition at 217 K, the spectra show insulating behavior on both sides of the transition (Fig. 2d), which is therefore an insulator-to-insulator transition. In the following, we present experimental evidence for distinct motives behind the two insulating states, which suggests that the I phase is, in fact, a Mott insulator.

The temperature evolution of flat band is shown in Fig. 3. The CDW band gap opens and the flat band forms at $E_F$ in the NC-CDW and T-CDW phases. Its bandwidth is narrow due to the large scale of the hexagonal SD structure. The flat band opens an energy gap and shifts from $E_F$ to −170 meV at 225 K in the I phase. In the C-CDW phase, the flat band shifts to around −200 meV and becomes more dispersive. Meanwhile, there is an energy spread of the flat band towards $E_F$, which is consistent with the peak broadening observed in Fig. 2d, e. Due to the surface sensitivity of ARPES technique, $k_z$ is not conserved during the photoemission process[28]. The ARPES spectra is thus a superposition of the bulk bands at different $k_z$'s. If the band has moderate band dispersion along $k_z$ direction, its ARPES spectra is broadened. Such $k_z$ broadening effect explains the energy spread of the flat band, which also indicates that the $k_z$ band dispersion reconstructs significantly in the C-CDW phase.

To determine the band dispersion of the flat band along the $k_z$ direction, we measured the photon energy-dependent data in the

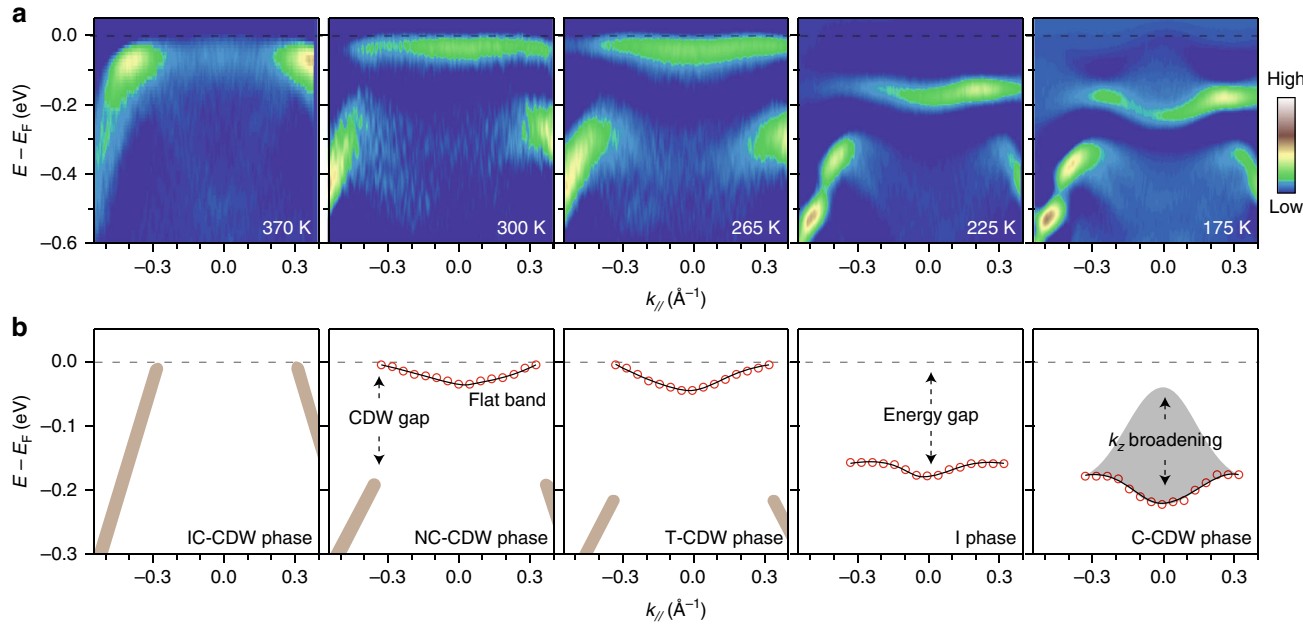

**Fig. 3 Evolution of the flat band crossing different electronic phases. a** The second derivative images of the energy–momentum cuts taken along the Γ–M direction in different electronic phases. **b** The schematic illustration of band evolution crossing different phases. Red circles show the fitted peak maxima from the data in **a**. Black lines guides the eyes to the flat band dispersions. Brown thick lines illustrate the bands at high binding energy.

I and C-CDW phases. The results are compared in Fig. 4. The flat band is gapped at 225 K. The weak photon energy dependence indicates the two dimensionality of the flat band in the I phase. However, when entering the C-CDW phase, the data become strongly photon energy dependent. The band positions shift from $-230$ to $-90$ meV periodically. This shows that the bandwidth along the $k_z$ direction is around 140 meV in the C-CDW phase, which is well consistent with the $k_z$ broadening effect observed in Fig. 3. The photon energy-dependent data confirm that the $k_z$ band dispersion reconstructs strongly at the I to C-CDW phase transition. More intriguingly, the periodicity of the $k_z$ dispersion is around $2\pi/2c$ in the C-CDW phase, which indicates the existence of an interlayer dimerization.

**Temperature dependence of lattice structure**. We then performed the single-crystal XRD measurement along the $(0, 0, L)$ direction to confirm the existence of interlayer stacking order. Indeed, in addition to the regular integer-indexed Bragg peaks (Fig. 5a), a set of half-integer-indexed reflections are observed at 120 K. The fact that the $(0, 0, 7/2)$ reflection is about twice more intense than the $(0, 0, 5/2)$ reflection, i.e., with the intensity roughly proportional to $|\mathbf{Q}|^2$, suggests that these additional reflections are caused by $c$-axis displacive structural modulations that double the unit cell along $c$ (Fig. 5c). The appearance of half-integer reflections in the C-CDW phase is a direct evidence for interlayer dimerization, which has to be caused by, and provide feedback to, interlayer coupling in the electronic structure. It is therefore no surprise that these half-integer reflections occur at 212 K which coincides with the enhancement of $k_z$ band dispersion observed by ARPES (Figs. 4 and 5b).

**Discussion**
For the C-CDW phase, its intra-layer CDW structure consists of SD clusters with the $p(\sqrt{13} \times \sqrt{13})R13.9°$ periodicity. This has been well studied by previous diffraction experiments and also confirmed by the ARPES mapping in Fig. 1b. Here, our XRD and photon energy-dependent ARPES data show that, in addition to the intra-layer CDW, the system forms an interlayer stacking

order along the $c$ direction. While the $2c$ periodicity could be attributed to an inter-layer dimerization, the stacking configuration between each bilayers remains unknown. Theoretical calculations show that the energy differences between different configurations are small[19,20]. Therefore, different stacking configurations could coexist, and the layers could stack randomly along the $c$ direction. This is consistent with our XRD data. The weak intensity of the half-integer diffraction peaks and the broadening of the CDW diffraction peak (Supplementary Fig. 5) suggest that the interlayer stacking shows certain randomness in the C-CDW phase. Therefore, to determine the stacking configuration experimentally, layer-resolved XRD studies are needed. Nevertheless, indirect evidences could be found by comparing the ARPES data with theoretical band calculations. Our result is more consistent with an alternating stacking configuration, where the predicted band structure is a small-gap band insulator[19,20].

Both ARPES and XRD data show the existence of interlayer stacking order in the C-CDW phase. This makes the low temperature ground state of $1T$-TaS$_2$ a band insulator with two electrons fully occupied in the flat band near $E_F$. The next question is then the origin of the I phase where the stacking order vanishes. Apparently, the I phase to C-CDW phase transition is an insulator-to insulator transition. Therefore, it cannot be explained by a simple interlayer Peierls CDW transition that is driven by the electronic instability at the Fermi surface. Electronic correlation need be considered. In a bilayer Hubbard band model, considering both the on-site Coulomb repulsion and interlayer hopping, theories show that the electronic system can transit from a Mott insulator to a band insulator when the interlayer hopping increase above a critical value[29,30]. The electronic states redistribute without closing the energy gap, resulting in an insulator-to-insulator transition. This naturally explains the C-CDW phase to I phase transition observed here (Fig. 5d, e). The I phase is a two-dimensional Mott insulator originated from the on-site Coulomb repulsion ($U$). When entering the C-CDW phase, the interlayer hopping ($t_\perp$) as characterized by the $k_z$ bandwidth is strongly enhanced. As a result, the system transits from a Mott insulator into a band insulator. Consistently, the

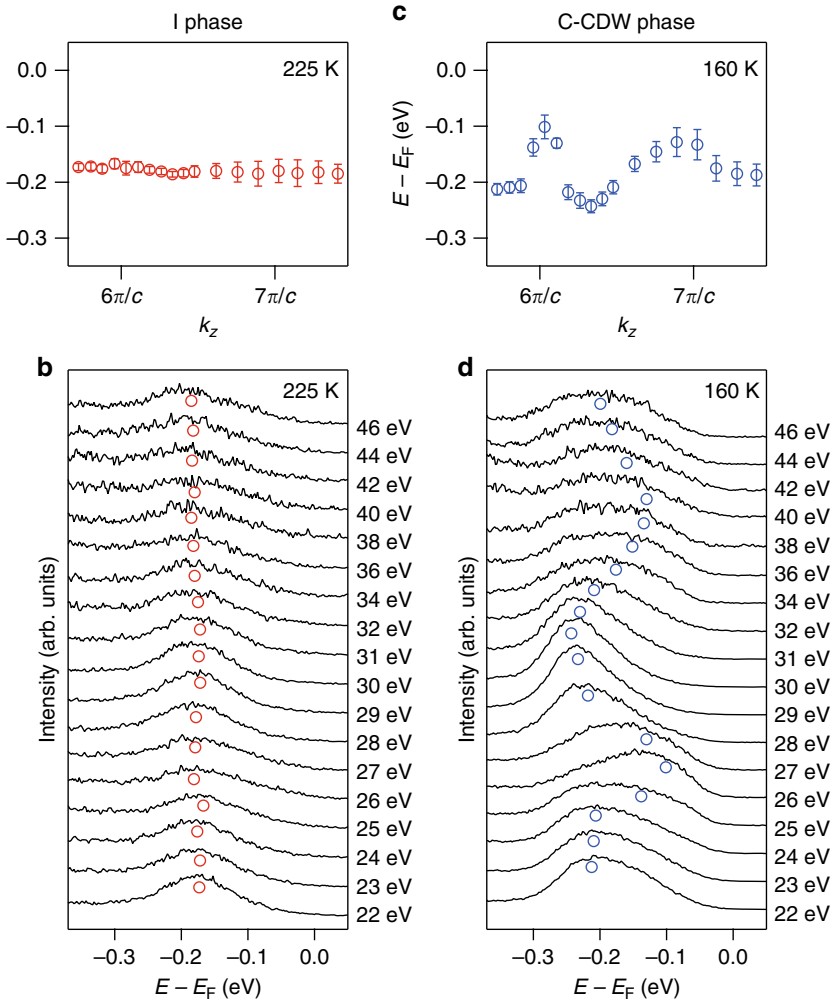

**Fig. 4 Band dispersions along the $k_z$ direction taken in the I and C-CDW phases. a** $k_z$ dependence of the band positions in the I phase at 225 K. $k_z$ is calculated using the inner potential $V_O = 17.5$ eV. **b** Photon energy dependence of EDCs taken at Γ at 225 K. **c** and **d** are the same as **a** and **b** but taken in the C-CDW phase at 160 K. Red and blue circles are the fitted band positions. Error bars are estimated from peak fittings and defined as s.d.

energy scale of the in-plane hopping ($t_{//}$) is around 100 meV as determined by the in-plane bandwidth of the flat band (Fig. 3). The Mott gap between the lower and upper Hubbard band is around 400 meV[13,18,25], representing the energy scale of $U$. The energy scale of $t_\perp$ is around 140 meV, which is the $k_z$ bandwidth of the flat band (Fig. 4). All these experimentally determined parameters are consistent with the theoretical parameters where the band-insulator to Mott-insulator transition could realize[29,30].

It is then intriguing to understand the enhancement of $t_\perp$ cross the I to C-CDW phase transition. In fact, the reconstruction of electronic structure along $c$ direction is so strong that upon entering the I phase from the C-CDW phase, the distance between adjacent TaS$_2$ layers undergoes an appreciable sudden decrease from 5.928 to 5.902 Å as seen from the (0, 0, 4) reflection (Fig. 5b). At first glance, the larger interlayer distance in the C-CDW phase compared to the I phase is at variance with the larger $t_\perp$ (hence stronger $k_z$ dispersion) in the C-CDW phase. This unexpected correlation between $t_\perp$ and interlayer distance requires further theoretical understanding. One possible explanation is that the complete in-plane commensurability of CDW in the C-CDW phase turns on additional hopping channels between nearby TaS$_2$ layers, which may further affect the average interlayer distance.

The phase transition at 233 K and the I phase have not been observed in both the resistivity and XRD data. Considering that APRES is more sensitive to the electronic state near the sample surface, we could explain this controversy using two different scenarios. On one hand, the I phase is a surface state that does not exist in the bulk layers. This resembles the Mott insulating phase that was observed at the surface of 1T-TaSe$_2$[31,32]. However, this scenario contradicts with some of our observations. For example, the penetration depth of ARPES is normally around or over two layers. If the surface and bulk electronic states are different, we should observe two different sets of bands representing the surface and bulk, respectively. However, our data only show one set of bands, which suggests that the I phase extends along $c$ direction for at least two TaS$_2$ layers. Furthermore, the phase transition at 217 K observed by ARPES is well consistent with the C-CDW transition temperature determined by transport and XRD measurements. The photon energy-dependent data also clearly show the $k_z$ dispersion of bands, whose periodicity is well consistent with the bulk lattice parameter. All these results suggest that the ARPES data reflect the bulk electronic properties of 1T-TaS$_2$. On the other hand, the I phase exists in bulk layers, but only emerges in certain layers which are phase separated from the other layers. Therefore, the I phase cannot be picked up by the resistivity and

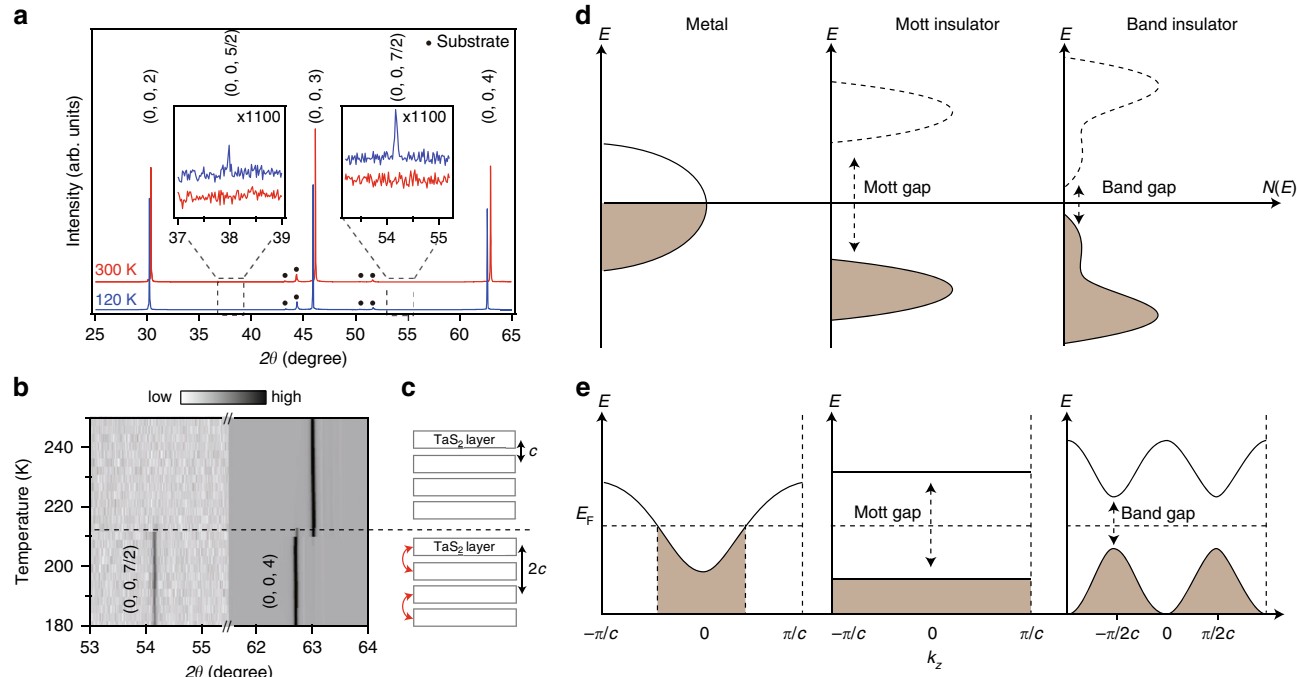

**Fig. 5 Temperature-dependent XRD study and schematic depictions of band evolution. a** XRD intensity of 1$T$-TaS$_2$ measured along $c$ axis at 120 and 300 K. Inset panels show the emergence of (0, 0, 5/2) and (0, 0, 7/2) peaks using 1100 times expanded scales. All the other peaks are from the polycrystalline substrate (Supplementary Fig. 4). **b** Temperature-dependence of the (0, 0, 7/2) and (0, 0, 4) peaks upon heating. **c** Schematic illustration of the interlayer dimerization. **d** Schematic illustration of the density of state distributions in different electronic phases. **e** Schematic illustration of the band dispersions along $k_z$ direction in different electronic phases.

XRD measurements. Under this scenario, the observation of I phase by ARPES should be highly dependent on the cleaved surface. However, this is not the case. The high reproducibility of our ARPES data seems to suggest that the I phase may have a high probability of occurrence near the sample surface.

To examine whether the I phase exists in the bulk or not, layer-resolved transport and XRD measurements are required. The resistivity measurements on thin 1$T$-TaS$_2$ films show that the transition becomes broad with the decreasing of sample thickness[33–35], which may suggest the existence of a phase separation. Moreover, the system remains insulating in an ultrathin film, while the C-CDW phase is fully suppressed[33–35]. This is consistent with our results that the interlayer hopping is important for the C-CDW phase. In an ultrathin film where the on-site Coulomb repulsion plays a more dominating role than the interlayer hopping, the insulating property may originate from the intermediate Mott insulating phase observed here.

In summary, we believe that the competition between $U$ (within SD) and $t_\perp$ (between neighboring TaS$_2$ layers) is responsible for the complex insulating phases in 1$T$-TaS$_2$. While the C-CDW phase is a band insulator featuring strong interlayer hoping and dimerization, the I phase is likely a two-dimensional Mott insulator. Although we show that the low-temperature C-CDW phase is not a Mott insulator, the $S = 1/2$ degrees of freedom could still be realized in the I phase. Further experimental studies are required to characterize possible magnetism in the I phase and verify its occurrence in bulk layers. Meanwhile, if the stacking order in the C-CDW phase can be suppressed, such as by chemical doping or reducing dimensionality, quantum magnetism may still be realized down to the lowest temperature in 1$T$-TaS$_2$.

## Methods

**Sample growth**. High-quality single crystals of 1$T$-TaS$_2$ were synthesized using chemical vapor transport (CVT) method. By mixing the appropriate ratio of Ta

powder and S pieces (2% excess) well, the compound was sealed in a quartz tube with ICl$_3$ as transport agent. The quartz tube was put in the two-zone furnace with thermal gradient between 750 and 850 °C for 120 h, and then quenched in water. The refinement of power XRD data confirms the phase purity of our sample and the trigonal crystal structure of 1$T$-TaS$_2$ with the $P$-3$m$1 space group with the lattice parameters $a = b = 3.366$ Å, $c = 5.898$ Å (Supplementary Fig. 6).

**Resistivity measurements**. Resistivity data were measured in physical property measurement system (PPMS, Quantum Design, Inc.) utilizing standard four-probe method. The heating and cooling rates are 3 K min$^{-1}$.

**ARPES measurements**. ARPES measurements were performed at Peking University using a DA30L analyzer and a helium discharging lamp. The photon energy of helium lamp is 21.2 eV. Photon energy-dependent measurement was performed at the BL13U beamline in National Synchrotron Radiation Laboratory (NSRL). The overall energy resolution was ~12 meV and the angular resolution was ~0.3°. The crystals were cleaved in situ and measured in vacuum with a base pressure better than $6 \times 10^{-11}$ mbar. The $E_F$ for the samples were referenced to that of a gold crystal attached onto the sample holder by Ag epoxy. For the measurements in a heating process, the samples were first cooled down from 300 to 80 K with a rapid fall of temperature (about 20 K min$^{-1}$). After adequate cooling for about 10 min at 80 K, the samples were cleaved and heated up to 190 K with a rate about 1.5 K min$^{-1}$; The samples were then heated up to 300 K slowly with a rate about 0.23 K min$^{-1}$. For the measurement in a cooling process, The sample was cleaved at 300 K directly and cooled down to 210 K with a rate about 2 K min$^{-1}$; The sample was then cooled down to 160 K slowly with a rate about 0.5 K min$^{-1}$. ARPES data were collected during the cooling and heating processes. More experimental details could be found in Supplementary Table 1.

**XRD measurements**. XRD data were recorded on a Bruker D8 diffractometer using Cu K$\alpha$ radiation ($\lambda = 1.5418$ Å). We mounted a high-quality single crystal of 1$T$-TaS$_2$ in a liquid nitrogen cryostat sitting in an Euler cradle to measure the $c$ axis at different temperatures.

The sample was cooled down until 93 K with a rapid fall of temperature (about 10 K min$^{-1}$) from room temperature naturally. After adequate cooling for about 1 h at 120 K, we heated the sample from 120 to 170 K at 3 K min$^{-1}$. We then heated the sample to 370 K with a rate about 0.16 K min$^{-1}$. XRD data were collected during the heating process.

**Experimental reproducibility**. The ARPES and resistivity measurements were repeated on different samples. The observations are reproducible in all measured samples with a small variable of transition temperatures (Supplementary Figs. 1–3). The resistivity data in Fig. 1a were taken on sample #7. The ARPES data in Fig. 2a, b were taken on sample #6. The ARPES data in Figs. 2d, f and 3 were taken on sample #1. The XRD data were taken on sample #13. The ARPES data taken at 370 K in Figs. 1 and 3 were taken on sample #14. Photon energy-dependent data in Fig. 4 was taken on sample #15.

## Data availability

The authors declare that the data supporting the findings of this study are available within the article and its Supplementary Information. All raw data are available from the corresponding author upon reasonable request.

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

## Acknowledgements

We gratefully thank Y. Zhang and Z. Liu for stimulating discussions. We gratefully thank S.T. Cui for his help on the ARPES experiment at NSRL. This work is supported by the National Natural Science Foundation of China (NSFC) (Grant No. 11888101), by the National Key Research and Development Program of China (Grant Nos. 2018YFA0305602 and 2016YFA0301003), and by the NSFC (Grant Nos. 11574004 and 91421107).

## Author contributions

Y.Z. conceived and instructed the project. W.L.Y. synthesized the single crystals. Y.L. supported the sample synthetization. Y.D.W. took the ARPES and XRD measurements with the contribution of Z.M.X., T.T.H., Z.G.W., L.C. and C.C. Y.D.W. and Y.Z. analyzed the data and wrote the paper with the input from all authors.

## Competing interests

The authors declare no competing interests.
