## [Peer Review File · Nature Communications]

Reviewers' comments:

Reviewer #1 (Remarks to the Author):

The manuscript reports results of temperature-dependent electrical resistance, ARPES, and XRD measurements of the layered CDW reference material 1T-TaS₂. The focus is on the temperature region where the pristine bulk material changes, upon heating, from a commensurate (C) CDW phase over a triclinic (T) CDW state to a nearly commensurate (NC), domain-like CDW phase. The major finding is a qualitative change in the ARPES spectral weight distribution and XRD intensity scan at an intermediate temperature (around 217 K) preceding the C-T CDW transition. Below this transition temperature, evidence is reported for an electronically gapped state with non-negligible band dispersion and a doubled unit cell in the z direction. In a small interval above the transition temperature (217-233 K), the band structure still exhibits a gap, but the dispersion appears more 2D-like and c-axis doubling is absent. The observation is interpreted as a transition from a Peierls insulating ground state to a novel intermediate Mott insulating state.

The study addresses an important open problem for 1T-TaS₂, i.e., the nature of its ground state. A flurry of experimental and theoretical activities has recently looked into this question, inspired by the finding that interlayer coupling, specifically CDW stacking, can play an important role in determining the electronic structure at the Fermi level. A common view in the field currently seems to be that a single S-Ta-S layer is a Mott insulator, while the bulk crystal is a Peierls insulator due to CDW stacking effects, and the surface of the bulk maybe somewhere in between. But clear experimental verification is still lacking so that the results of the present study are, in principle, timely, relevant, and interesting.

Unfortunately, however, there are a number of major issues that prevent me from concluding that the presented results can contribute to a better understanding of the phase transition behavior of the material and its ground state:

(i) There is a significant inconsistency in the phase transition behavior observed with the three different techniques used. The measured resistance displays the known behavior: NC-C transition at 180 K upon cooling and C-T transition at 220 K upon heating. The ARPES data, by contrast, show a NC-C transition upon cooling at 193 K (Fig. 2c) and the C-I transition upon heating at 217 K (Figs. 2e and 2f), whereas this latter transition is observed in the XRD data at 212 K (Fig. 4b). These differences call into question the quality of the samples and the reproducibility of the results. The authors explicitly state that "the temperature-dependent data are well reproducible in many different samples", but they do not show the data.

(ii) The ARPES evidence for non-negligible k_z dispersion in the low-temperature C phase is only indirect, provided by the supposedly k_z smearing-induced spectral weight shadow reflecting the projected band dispersion along the z direction (Fig. 3a, rightmost panel, taken at 175 K). What is disturbing here is that a different data normalization has apparently been applied to this ARPES intensity map compared to the one next to it taken in the intermediate "I" phase at 225 K (this can be seen in the photoemission intensity above the Fermi level). In order to convincingly establish the presence or absence of k_z dispersion in the C and I phase, respectively, photon energy-dependent measurements are required.

(iii) Although the XRD evidence for c-axis doubling in the low-temperature C phase appears to be more direct, as given by the (0, 0, 5/2) and (0, 0, 7/2) superstructure reflections in Fig. 4b, the origin of these diffraction peaks, their intensities, and the underlying CDW stacking order remain completely unclear. This potentially novel result is also not put into the context of the results of previous structural determinations of the C phase. A convincing structural analysis of the transition to the

intermediate phase is needed.

(iv) A key difference between the applied experimental techniques is neither mentioned nor discussed in the manuscript: transport and XRD measurements are bulk sensitive, while ARPES is highly surface sensitive. So, an important question is to what extent the reported ARPES results are representative of bulk behavior. Considering the differences in transition temperatures and the absence of the 233 K transition from the reported bulk data, it may well be that the intermediate phase seen in ARPES is a surface effect, similar to the surface Mott transition that is known to occur in the sister material 1T-TaSe₂. That is, the intermediate phase would explicitly not reflect "an intrinsic property of 1T-TaS₂".

Some minor points are the following:

(v) The c lattice-parameter change occurring at 212 K should be quantified.

(vi) Heating and cooling rates for the ARPES and XRD measurements should be stated.

(vii) Experimental details of the transport measurements should be given.

In conclusion, while the combined ARPES and XRD results are in principle interesting, the experimental evidence provided is mostly indirect and there is a lack of technical rigor so that the central conclusion ("transition from band insulator to Mott insulator at intermediate temperature") is not substantiated by the data. Furthermore, the discussion and explanation of the results remains at a purely qualitative level, whereas quantitative insight is needed to make progress in better understanding this intriguing material. The manuscript in its present form cannot be recommended for publication. A major revision is needed.

Reviewer #2 (Remarks to the Author):

The authors of the article "Band insulator to Mott insulator transition in 1T-TaS₂" have performed resistivity, ARPES and X-ray diffraction measurement of the transition metal dichalcogenide 1T-TaS₂. Their data convincingly show that a dimerization of the unit cell in the stacking direction takes place at metal insulator transition. These data confirm important results that have been already published in Ref.8 and Ref.9 of the article. In addition, the authors discovered the presence of a new phase that shows up at 217 K while heating up the sample. After reading the article, we decided to perform the experiment our self and we confirm the existence of such new phase.

This article is very interesting, well written and based on reproducible, high quality data. I have no doubt that will generate the interest of broad community. On the other hand, minor comments should be taken into account in order to improve the manuscript:

1) The authors propose that the new phase is a Mott insulator. However, the resistivity shows a jump to down to the metallic value when heating above 217 K. The authors should explain this apparent incongruity.

2) It would be very valuable if the authors could estimate the magnitude of displacement related to the dimerization.

3) The authors have cited in Ref.23 a time resolved ARPES experiment on 1T-TaS₂ but they overlooked to cite the first work on this subject (L. Perfetti, Physical review letters 97, 067402 (2006)). This negligence must be corrected.

Reviewer #3 (Remarks to the Author):

The authors performed temperature-dependent ARPES and XRD measurements, giving a band insulating ground state with interlayer dimerization of bulk 1T-TaS₂ single crystal and existence of an intermediate Mott insulating state. This layered compound with simple crystal structure provides a great platform to investigate its complex electronic phase transitions (multi-type CDWs, superconductivity and so on) and the physical origins, from bulk crystals down to few-layer devices. Recent theoretical and STM studies on few-layer samples emphasize the importance of interlayer stacking order. I would suggest the following comments should be addressed before consideration.

- 1) As stated in the end of this manuscript, it is intriguing that no transport anomaly responds to the observed intermediate Mott insulating state. Will that be observable in thin sample? What is the thickness of the sample measured in this study? Since the contribution of interlayer interaction would be more significant in thin sample, I would suggest the authors to compare the current results with that on thin sample.
- 2) line 105: as noted, the temperature-dependent data are well reproducible in many different samples. I would suggest the authors plot all the data in comparison, at least presenting it in supplementary material as well as the differences between used samples.
- 3) Figure 4(a): I would suggest that the intensity (y-axis) should be better presented in log scale since the (00l) peaks are too strong. Some other peaks other than (00l) and (00l/2) should also be addressed.
- 4) The quality of 1T phase single crystals strongly depends on the quenching process and growing temperatures. More details of the quality of the single crystals should be addressed, such as SEM image, EDX result (usually S-deficiency exists), powder XRD and refinement and so on.

The reviewers' comments are in italic, and our reply follows.

Reviewer #1 (Remarks to the Author):

The manuscript reports results of temperature-dependent electrical resistance, ARPES, and XRD measurements of the layered CDW reference material 1T-TaS₂. The focus is on the temperature region where the pristine bulk material changes, upon heating, from a commensurate (C) CDW phase over a triclinic (T) CDW state to a nearly commensurate (NC), domain-like CDW phase. The major finding is a qualitative change in the ARPES spectral weight distribution and XRD intensity scan at an intermediate temperature (around 217 K) preceding the C-T CDW transition. Below this transition temperature, evidence is reported for an electronically gapped state with non-negligible band dispersion and a doubled unit cell in the z direction. In a small interval above the transition temperature (217-233 K), the band structure still exhibits a gap, but the dispersion appears more 2D-like and c-axis doubling is absent. The observation is interpreted as a transition from a Peierls insulating ground state to a novel intermediate Mott insulating state.

The study addresses an important open problem for 1T-TaS₂, i.e., the nature of its ground state. A flurry of experimental and theoretical activities has recently looked into this question, inspired by the finding that interlayer coupling, specifically CDW stacking, can play an important role in determining the electronic structure at the Fermi level. A common view in the field currently seems to be that a single S-Ta-S layer is a Mott insulator, while the bulk crystal is a Peierls insulator due to CDW stacking effects, and the surface of the bulk maybe somewhere in between. But clear experimental verification is still lacking so that the results of the present study are, in principle, timely, relevant, and interesting.

Unfortunately, however, there are a number of major issues that prevent me from concluding that the presented results can contribute to a better understanding of the phase transition behavior of the material and its ground state:

(i) There is a significant inconsistency in the phase transition behavior observed with the three different techniques used. The measured resistance displays the known behavior: NC-C transition at 180 K upon cooling and C-T transition at 220 K upon heating. The ARPES data, by contrast, show a NC-C transition upon cooling at 193 K (Fig. 2c) and the C-I transition upon heating at 217 K (Figs. 2e and 2f), whereas this latter transition is observed in the XRD data at 212 K (Fig. 4b). These differences call into question the quality of the samples and the reproducibility of the results. The authors explicitly state that "the temperature-dependent data are well reproducible in many different samples", but they do not show the data.

We added ARPES and transport data in the supplementary material (Supplementary Figures 1-3). The ARPES data in Supplementary Figure 1 declare the repetitiveness of our observation well. The intermediate state and the two phase transitions could be observed in all measured samples.

We summarized the phase transition temperatures determined by ARPES, resistivity, and XRD measurements in Supplementary Figure 3. There is a small variation of transition temperatures among different samples, but the temperature deviation is less than 10K. For technique reasons, we cannot measure the same sample using the transport, AREPS and XRD techniques. Therefore, it is normal that the transition temperatures measured by different techniques are different. Such transition temperature difference is under the standard deviation and can be explained by a small sample inhomogeneity. This does not invalidate the conclusions of our manuscript.

We explained this point in the revised manuscript. We would also like to point out that, in referee #2's report, referee #2 confirmed our observation after performing experiments themselves. Based on all these results, we believe our observations are solid and reproducible.

(ii) The ARPES evidence for non-negligible k_z dispersion in the low-temperature C phase is only indirect, provided by the supposedly k_z smearing-induced spectral weight shadow reflecting the projected band dispersion along the z direction (Fig. 3a, rightmost panel, taken at 175 K). What is disturbing here is that a different data normalization has apparently been applied to this ARPES intensity map compared to the one next to it taken in the intermediate "I" phase at 225 K (this can be seen in the photoemission intensity above the Fermi level). In order to convincingly establish the presence or absence of k_z dispersion in the C and I phase, respectively, photon energy-dependent measurements are required.

Following the referee's suggestion, we measured the band dispersions along the k_z direction in both I and C-CDW phase using synchrotron-based ARPES. The data are shown in Fig.4 in the revised manuscript. The new data provide direct evidences, which are consistent with our temperature-dependent data and strengthen our conclusions.

The flat band is gapped and shows weak photon energy dependence in the intermediate phase at 225K. This indicates the two dimensionality of the flat band in the I phase. However, when entering the C-CDW phase, the data become strongly photon energy dependent. The band positions shift from -230meV to -90meV periodically. First, the measured bandwidth along k_z direction is around 140 meV in the C-CDW phase, which is well consistent with the k_z broadening effect observed by the helium lamp (21.2 eV). Second, the period of the k_z dispersion is around $2\pi/2c$, which suggests the existence of interlayer dimerization. Third, the weak k_z dispersion and large energy gap observed in the I phase suggest that the I phase is a Mott insulating phase where the Column repulsion play a more dominating role than the inter-layer hopping.

(iii) Although the XRD evidence for c-axis doubling in the low-temperature C phase appears to be more direct, as given by the (0, 0, 5/2) and (0, 0, 7/2) superstructure reflections in Fig. 4b, the origin of these diffraction peaks, their intensities, and the underlying CDW stacking order remain completely unclear. This potentially novel result is also not put into the context of the results of previous structural determinations of the C phase. A convincing structural analysis of the transition to the intermediate phase is needed.

The structure of the C-CDW phase can be divided into three tiers: the intra-layer CDW, inter-layer dimerization, and inter-layer stacking configuration. First, the previous STM and XRD studies show that the intra-layer CDW is a $\sqrt{13} \times \sqrt{13}$ star-of-David structure in the C-CDW phase. This has been also confirmed by our ARPES mapping in Fig.1. Second, our XRD and photo-energy-dependent ARPES data show that adjacent layers form an inter-layer dimerization with a $2\pi/2c$ period. Then, the only unknown structure is the interlayer stacking configuration. There are three proposed configurations, on-top stacking, alternating stacking and nonalternating stacking (PRB 98,195134, PRL 122,106404). Theoretically, it has been proposed that the energy differences between different configurations are small. Therefore, different stacking configurations could coexist, and the layers could stack randomly along the c direction. We have measured the temperature dependence of the $\sqrt{13} \times \sqrt{13}$ diffraction peak using XRD. The results are plotted in Supplementary Figure 5. The diffraction peak broadens remarkably below 212 K. Together with the weak intensity of the half-integer diffraction peaks, our XRD data is consistent with the theoretical prediction that the interlayer stacking shows certain randomness in the C-CDW phase. To fully determine the stacking configuration experimentally, layer-resolved XRD measurement is required.

Although the stacking configuration might be randomness and is difficult to be determined experimentally. Indirect evidences could be found by comparing our ARPES results with theoretical band calculations. Band calculations show that for the on-top and nonalternating stacking configurations, the system is metallic with one band crossing the Fermi energy. This is inconsistent with our observation. Our results are more consistent with an alternating stacking configuration where the predicted band structure is a small-gap band insulator.

We rewrote the discussion part to explain this point explicitly in the revised manuscript.

(iv) A key difference between the applied experimental techniques is neither mentioned nor discussed in the manuscript: transport and XRD measurements are bulk sensitive, while ARPES is highly surface sensitive. So, an important question is to what extent the reported ARPES results are representative of bulk behavior. Considering the differences in transition temperatures and the

absence of the 233 K transition from the reported bulk data, it may well be that the intermediate phase seen in ARPES is a surface effect, similar to the surface Mott transition that is known to occur in the sister material 1T-TaSe₂. That is, the intermediate phase would explicitly not reflect “an intrinsic property of 1T-TaS₂”.

As mentioned by the referee, the 233K phase transition and the insulating intermediate phase have not been observed in the resistivity and XRD data. Considering that APRES is more sensitivity to the electronic state near the sample surface, we could explain this controversy using two different scenarios.

First, the I phase could be a surface state. It only exists at the surface layer and doesn't exist in the bulk layers. However, this scenario contradicts with some of our observations. For example, with 21.2 eV photons, the penetration depth of ARPES is normally around or over 2 layers. If the surface and bulk electronic states are different, we should observe two different sets of bands representing the surface and bulk respectively. However, our data only show one set of bands, which suggests that the extension of I phase along c direction is at least two TaS₂ layers. Furthermore, the first phase transition observed by ARPES is around 217K, which is well consistent with the C-CDW transition temperature determined by transport and XRD measurements. Our photon energy dependent data clearly observed the k_z dispersion of bands, whose period is well consistent with the bulk lattice parameter. All these results suggest that the ARPES data reflect the bulk properties of 1T-TaS₂.

Another scenario is that the system is inhomogeneous near the phase transition and shows phase separation among different layers. The I phase only exists in certain layers and therefore cannot be picked up by the resistivity and XRD measurements. Under this scenario, the observation of I phase by ARPES should be highly dependent on the cleaved surface. However, this is not the case. The high reproducibility of our ARPES data seems to suggest that the I phase may have a high probability of occurrence near the sample surface.

While our results undoubtedly show the existence of I phase in the few topmost layers of 1T-TaS₂. To determine whether the I phase exists in the bulk or not, layer-resolved transport and XRD measurements are required. We agree with the referee that it is an intriguing open question. But, we think this is beyond the scope of our manuscript. The main focus of our manuscript is the discovery of intermediate Mott phase and its transition to a band insulator with interlayer dimerization. We think this captures the most important physics of 1T-TaS₂, which explains its rich phases and electronic properties. We hope our results will stimulate following researches working on this unanswered question in this intriguing material.

We rewrote the discussion part to explain this point explicitly in the revised manuscript.

Some minor points are the following:

(v) The c lattice-parameter change occurring at 212 K should be quantified.

We added the c lattice-parameters in the revised manuscript. The c lattice-parameter is 5.902Å at high temperatures and 5.928Å in the C-CDW phase.

(vi) Heating and cooling rates for the ARPES and XRD measurements should be stated.

For the ARPES measurements in a heating process, the samples were cooled down until 80 K with a rapid fall of temperature (about 20 K/min) from room temperature (RT, 300 K) naturally. After adequate cooling for about 10 min at the lowest temperature, the samples were heated to 160 K nearby with a rate about 1.5 ~ 5 K/min. The samples were then heated up slowly (0.22 ~ 0.4 K/min) to witness the bands evolution near the phase transitions. ARPES data were collected during the heating process. For the ARPES measurement in a cooling process, The sample was cleaved at 300 K and cooled down to 210 K with a rate about 2 K/min; The sample was then cooled down to 160 K slowly with a rate about 0.5 K/min. ARPES data were collected during the cooling process. More experimental details are shown in Tab. S1.

For the XRD measurement, the sample was cooled down until 93 K with a rapid fall of temperature (about 10 K/min) from RT naturally. After adequate cooling for about 1 hour at 120 K, we heated the sample to 170 K with a rate of 3 K/min. We then heated the sample to 370 K with a rate of 0.16 K/min. The XRD data were collected during the heating process.

We added the experimental details in the revised manuscript and supplementary material.

(vii) Experimental details of the transport measurements should be given.

The transport data were measured in Physical Property Measurement System (PPMS, Quantum Design, Inc.) utilizing the standard four-probe method. The heating and cooling rates are 3K/min.

We added the experimental details for the transport measurements in the revised manuscript.

In conclusion, while the combined ARPES and XRD results are in principle interesting, the experimental evidence provided is mostly indirect and there is a lack of technical rigor so that the central conclusion (“transition from band insulator to Mott insulator at intermediate temperature”) is not substantiated by the data. Furthermore, the discussion and explanation of the

results remains at a purely qualitative level, whereas quantitative insight is needed to make progress in better understanding this intriguing material. The manuscript in its present form cannot be recommended for publication. A major revision is needed.

We thank the referee for thinking our results interesting. We have conducted the photon energy dependent experiment following referee's suggestion. The new data provide strong and direct evidences, which strengthen our conclusions. We hope the referee will agree that our conclusion is based on high quality and reproducible data.

The main focus of our paper is the discovery of intermediate Mott phase and its transition to a band insulator with interlayer dimerization. We agree with referee that the interlayer stacking configuration and the existence of I phase in bulk layers are intriguing open questions. However, it is technically difficult to provide definite answers to these questions. We rewrote the discussion part to explain all possible scenarios explicitly in the revised manuscript. We hope our results and discussions could stimulate the following theoretical and experimental researches focusing on these unanswered questions.

Our results do provide quantitative information, such as the energy scales of $t_{//}$, U , and t_{\perp} . These parameters are critical for constructing a correct theoretical mode of this intriguing material. The comparison between the experimental parameters and theories support our conclusions that the observed transition is a Mott insulator to band insulator transition.

We thank the referee's comments and suggestions, which helped us improving the paper significantly. We hope the referee would found the revised manuscript satisfactory.

Reviewer #2 (Remarks to the Author):

The authors of the article “Band insulator to Mott insulator transition in 1T-TaS₂” have performed resistivity, ARPES and X-ray diffraction measurement of the transition metal dichalcogenide 1T-TaS₂.

Their data convincingly show that a dimerization of the unit cell in the stacking direction takes place at metal insulator transition. These data confirm important results that have been already published in Ref.8 and Ref.9 of the article. In addition, the authors discovered the presence of a new phase that shows up at 217 K while heating up the sample. After reading the article, we decided to perform the experiment our self and we confirm the existence of such new phase.

This article is very interesting, well written and based on reproducible, high quality data. I have no doubt that will generate the interest of broad community. On the other hand, minor comments should be taken into account in order to improve the manuscript:

1) The authors propose that the new phase is a Mott insulator. However, the resistivity shows a jump to down to the metallic value when heating above 217 K. The authors should explain this apparent incongruity.

We thank the referee for his/her suggestion. We explained this point explicitly in the page 4 of our reply and also in the revised manuscript. The inconsistencies between ARPES and transport data question the existence of Mott phase in the bulk layers. While our data undoubtedly show the existence of intermediate Mott phase in the few topmost layers of 1T-TaS₂, to determine whether the Mott phase exists in bulk layers or not, the layer-resolved transport or XRD experiments are required. We hope our results could stimulate following theoretical and experimental researches to clarify this issue.

2) It would be very valuable if the authors could estimate the magnitude of displacement related to the dimerization.

We added the c lattice-parameters in the revised manuscript. The c lattice-parameter is 5.902Å at high temperatures and 5.928Å in the C-CDW phase.

3) The authors have cited in Ref.23 a time resolved ARPES experiment on 1T-TaS₂ but they overlooked to cite the first work on this subject (L. Perfetti, Physical review letters 97, 067402 (2006)). This negligence must be corrected.

We thank the referee for his/her correction. We added the related reference in the revised manuscript.

We thank the referee for his/her comments and suggestions, which helped us improve the paper significantly. We hope the referee would find the revised manuscript satisfactory.

Reviewer #3 (Remarks to the Author):

The authors performed temperature-dependent ARPES and XRD measurements, giving a band insulating ground state with interlayer dimerization of bulk 1T-TaS₂ single crystal and existence of an intermediate Mott insulating state. This layered compound with simple crystal structure provides a great platform to investigate its complex electronic phase transitions (multi-type CDWs, superconductivity and so on) and the physical origins, from bulk crystals down to few-layer devices. Recent theoretical and STM studies on few-layer samples emphasize the importance of interlayer stacking order. I would suggest the following comments should be addressed before consideration.

1) As stated in the end of this manuscript, it is intriguing that no transport anomaly responds to the observed intermediate Mott insulating state. Will that be observable in thin sample? What is the thickness of the sample measured in this study? Since the contribution of interlayer interaction would be more significant in thin sample, I would suggest the authors to compare the current results with that on thin sample.

We thank the referee for his/her suggestion. We explained this point explicitly in the page 4 of our reply and also in the revised manuscript. The inconsistencies between ARPES and transport data question the existence of Mott phase in the bulk layers. While our data undoubtedly show the existence of intermediate Mott phase in the few topmost layers of 1T-TaS₂, to determine whether the Mott phase exists in bulk layers or not, the layer-resolved transport or XRD experiments are required. The transport or XRD experiments taken on thin 1T-TaS₂ film would be very helpful.

The resistivity measurements on thin 1T-TaS₂ Films have been reported in many literatures [Sci. Rep 4, 7302 (2014), PNAS 112, 15054-15059 (2015), Sci. Adv. 1, e1500606 (2015), Nat. Nanotech. 10, 270-276 (2015)]. Despite of their inconsistencies, they share some common results. First, the transition becomes broad with the decreasing of sample thickness, which may suggest the existence of phase inhomogeneous. Second, the sharp increment of resistivity at 217K disappears in thin films, indicating a disappearance of C-CDW phase. This is consistent with our conclusion that the interlayer hopping is important for the C-CDW phase. Third, the system's ground state remains insulating in the ultrathin film whose thickness is down to 2nm. In such an ultra-thin film, the intra-layer interactions should play a more dominating role than the interlayer hopping. Therefore, the insulating phase observed in ultra-thin films could be attributed to the Mott phase observed here.

We explained this point in the revised manuscript. While the current results on thin film do not contradict our conclusions. More detailed experiments are required in order to locate the existence of Mott phase in bulk layers and characterize its properties.

2) line 105: as noted, the temperature-dependent data are well reproducible in many different samples. I would suggest the authors plot all the data in comparison, at least presenting it in supplementary material as well as the differences between used samples

We added ARPES and transport data in the supplementary material (Supplementary Figures 1-3). The ARPES data in Supplementary Figure 1 declare the repetitiveness of our observation well. The intermediate state and the two phase transitions could be observed in all measured samples.

3) Figure 4(a): I would suggest that the intensity (y-axis) should be better presented in log scale since the (00l) peaks are too strong. Some other peaks other than (00l) and (00l/2) should also be addressed.

We plot the data in linear scale to better illustrate the intensity difference between the (0, 0, L/2) peaks. We plotted the data in log-scale in Supplementary Figure 4 in supplementary material. All the other peaks come from the polycrystalline substrate and are temperature independent.

4) The quality of 1T phase single crystals strongly depends on the quenching process and growing temperatures. More details of the quality of the single crystals should be addressed, such as SEM image, EDX result (usually S-deficiency exists), powder XRD and refinement and so on.

We added the experimental details of our sample growth in the revised manuscript. To prove the high quality of our sample, we added the sample photo, power XRD and rocking curve data in Supplementary Figure 6 in the supplementary material. Firstly, the sample is large and shows mirror-like naturally cleaved surface. Secondly, the diffraction peaks are narrow. The FWHMs are 0.0185° and 0.0149° in the rocking curve and two-theta scan respectively. Thirdly, the power XRD data are consistent with the reported data showing a pure 1T phase of TaS₂. The refinement confirmed the phase purity and the trigonal crystal structure of 1T-TaS₂ with the *P-3m1* space group with lattice parameters $a = b = 3.366 \text{ \AA}$, $c = 5.898 \text{ \AA}$ (Supplementary Figure 6). Fourthly, our resistivity data is highly reproducible with a standard deviation of transition temperature less than 10K. Our resistivity data are well consistent with the data reported in many literatures. All these results indicate a high quality of our sample.

We would also like to point out that, in referee #2's report, referee #2 confirmed our observation after performing experiments themselves. We believe our experimental observations are solid and reproducible.

We thank the referee for his/her comments and suggestions, which helped us improving the paper significantly. We hope the referee would found the revised manuscript satisfactory.

REVIEWERS' COMMENTS:

Reviewer #1 (Remarks to the Author):

The authors have carefully addressed all points raised by the reviewers; I find their responses in principle convincing. Following the reviewers' request, the authors have added new photon energy-dependent ARPES data as well as ARPES, transport, and XRD results across 13 different samples. They have also extended their discussion to include structural aspects of the observed phases and the possible interpretation of the novel intermediate phase as a surface effect. The additions to the manuscript and supplementary information have improved the transparency of the analysis and the overall quality and consistency of the work and made it more quantitative; the added supplementary information has particularly increased trust in the reproducibility of the results. In my opinion, the quality and scientific content of the manuscript have improved to a level that justifies publication in Nature Communications. Yet, a careful proofreading of the manuscript and supplementary material is required.

Reviewer #2 (Remarks to the Author):

The authors included k_z dispersion of electronic states, making their arguments more convincing. They have replied satisfactorily to the comments of the referees except for one point. As stated by Referee 1, the new phase that they observe at the surface of 1T-TaS₂ bears strong resemblance to the one reported in 1T-TaSe₂ (L. Perfetti et. al. Physical review letters 90, 166401 (2003)). In the discussion section, the authors should compare the I phase of 1T-TaS₂ to the results published in the parent compound 1T-TaSe₂. I would recommend publication if the authors take into account this remark.

Luca Perfetti

Reviewer #3 (Remarks to the Author):

The authors have adequately addressed my original questions concerning the manuscript and I feel that the paper is now suitable for publication in Nature Communication.

The reviewers' comments are in italic, and our reply follows.

Reviewer #1 (Remarks to the Author):

The authors have carefully addressed all points raised by the reviewers; I find their responses in principle convincing. Following the reviewers' request, the authors have added new photon energy-dependent ARPES data as well as ARPES, transport, and XRD results across 13 different samples. They have also extended their discussion to include structural aspects of the observed phases and the possible interpretation of the novel intermediate phase as a surface effect. The additions to the manuscript and supplementary information have improved the transparency of the analysis and the overall quality and consistency of the work and made it more quantitative; the added supplementary information has particularly increased trust in the reproducibility of the results. In my opinion, the quality and scientific content of the manuscript have improved to a level that justifies publication in Nature Communications. Yet, a careful proofreading of the manuscript and supplementary material is required.

We thank the referee for his/her recommendation.

Reviewer #2 (Remarks to the Author):

The authors included k_z dispersion of electronic states, making their arguments more convincing. They have replied satisfactorily to the comments of the referees except for one point. As stated by Referee 1, the new phase that they observe at the surface of 1T-TaS₂ bare strong resemblance to the one reported in 1T-TaSe₂ (L. Perfetti et. al. Physical review letters 90, 166401 (2003)). In the discussion section, the authors should compare the I phase of 1T-TaS₂ to the results published in the parent compound 1T-TaSe₂. I would recommend publication if the authors take into account this remark.

Luca Perfetti

We thank the referee for his/her recommendation. We agree with referee that the I phase observed here resembles the Mott insulating phase that was observed at the surface of 1T-TaSe₂. Following the referee's suggestion, we included the above statement in the discussion section of the revised manuscript. We added two related references in the manuscript and referred to them in the manuscript.

Reviewer #3 (Remarks to the Author):

The authors have adequately addressed my original questions concerning the manuscript and I feel that the paper is now suitable for publication in Nature Communication.

We thank the referee for his/her recommendation.